# Severe Cranio-Cervical Stenosis in a Child with Saul-Wilson Syndrome: A Case Report

**DOI:** 10.3390/children9040532

**Published:** 2022-04-08

**Authors:** Nenad Koruga, Silvija Pušeljić, Višnja Tomac, Anamarija Soldo Koruga, Igor Marjanac, Borna Biljan, Krešimir Šantić, Ivana Lenz, Nora Pušeljić

**Affiliations:** 1Department of Neurosurgery, University Hospital Center Osijek, J. Huttlera 4, 31000 Osijek, Croatia; 2Faculty of Medicine, Josip Juraj Strossmayer University of Osijek, 31000 Osijek, Croatia; puseljic.silvija@kbo.hr (S.P.); visnja.tomac@yahoo.com (V.T.); anamarijasoldo@gmail.com (A.S.K.); igor.marjanac@gmail.com (I.M.); borna.biljan@gmail.com (B.B.); santic.kresimir@gmail.com (K.Š.); glavica.ivana@gmail.com (I.L.); nora.puseljic@gmail.com (N.P.); 3Pediatric Clinic, University Hospital Center Osijek, J. Huttlera 4, 31000 Osijek, Croatia; 4Department of Neurology, University Hospital Center Osijek, J. Huttlera 4, 31000 Osijek, Croatia

**Keywords:** skeletal dysplasia, microcephaly, coxa valga, brachydactyly

## Abstract

Introduction: Saul Wilson syndrome (SWS) is a rare congenital syndrome characterized by a variety of symptoms, mostly skeletal changes. Saul and Wilson were the first to describe children with extremely short stature and craniofacial dysmorphism. Case report: We present a case of a 15-years-old boy with clinical and radiological characteristics of SWS. Genetic examination identified a pathogenic heterozygous variant in the *COG4* gene. Magnetic resonance imaging revealed a critical stenosis of the cranio-cervical junction (CCJ) which required surgical treatment to attempt sufficient neurological decompression. The patient underwent decompression of CCJ under general anesthesia. There was no significant radiological and clinical improvement during the postoperative period. Conclusions: SWS is presented as an extremely rare congenital disease in children. The clinical condition of our patient confined surgical possibilities, therefore further treatment in such patients should be appropriately evaluated.

## 1. Introduction

Saul-Wilson syndrome (SWS) (OMIM#618150) is caused by recurrent de novo heterozygous missense mutations in the conserved oligomeric Golgi—*COG4* (606976) gene on chromosome 16q22, and represents a rare genetic autosomal dominant disorder typically caused by a de novo pathogenic variant. SWS is a skeletal dysplasia characterized by numerous skeletal disorders, such as profound short stature, clubfoot and short distal phalanges of fingers and toes. Besides skeletal features, patients usually have characteristic craniofacial features such as prominent forehead, prominent eyes and micrognathia, ocular disorders and hearing loss. Early development is usually delayed with normal cognition. The diagnosis of SWS is established in a proband with marked short stature, typical facial and skeletal features, and a heterozygous pathogenic variant in *COG4* identified by molecular genetic testing [1].

COG4 is a subunit of the COG complex and an important intermediary in transport of proteins between the Golgi apparatus to the endoplasmic reticulum. Mutations of the *COG4* gene result in altered proteins in the COG complex and change retrograde transport of proteins, which consequently results as SWS [2]. To date only two *COG4* variants have been reported; both resulted in a p.Gly516Arg missense change [1].

The authors present a rare case of a boy suffering from SWS complicated with critical stenosis of cranio-cervical junction (CCJ) and atlanto-axial dislocation.

### 1.1. Case Report

A 15-year-old boy suffering from SWS was born from the third pregnancy of nonconsaguine young and healthy parents as a hypotrophic infant with a perinatal risk factor which included intrauterine growth retardation and asphyxia. Complex phenotypic dysmorphic features were observed at birth. The phenotype of our patient includes multiple facial and skeletal disorders: midface retrusion, prominent forehead, frontal bossing, shallow orbits, downslanted palpebral fissures, narrow nasal bridge, large filtrum, micrognathia, small hands with short metacarpal bones and short distal phalanges of fingers with consequent brachydactyly, knee flexion contracture, pseudoarthrosis, calcaneovalgus deformity, thoracic scoliosis, lumbar hyperlordosis, coxa valga, skeletal dysplasia, pectus carinatum, generalized hypotonia, global developmental delay, cognitive impairment, complete lack of adipose tissue with consequent pseudohypertrophy of muscles, bilateral cataracts and progeroid facial appearance (Figure 1 and Figure 2). As a part of the syndrome, an extremely short stature, body weight of 15 kg and body height of 100 cm were observed.

Computed tomography (CT) scan of cranio-cervical junction revealed bifid arch of the atlas as a part of skeletal disorders. Additional preoperative magnetic resonance imaging (MRI) revealed a critical stenosis of cranio-cervical junction with concomitant myelopathy (Figure 3).

### 1.2. Genetic Exam

Karyotyping excluded chromosomopathies, and no change of clear clinical significance in chromosomal microarray analysis (CMA) was found. Whole exome sequencing (genomic DNA was extracted from buccal swab sample) identified a pathogenic heterozygous variant in *COG4* gene (16-70530270-C-T NM_015386.3:c.1546G>A (NP_056201.2:p.Gly516Arg). Amino acid changes identical to a pathogenic variant have been previously reported (ClinVarID: VCV000585271). The same variant was also reported as de novo in one or more affected individuals with a consistent phenotype from multiple, unrelated families.

### 1.3. Surgical Treatment and Early Follow-Up

The patient underwent surgical treatment under general anesthesia and in a prone position. The surgical approach was performed in the mediosagittal plane. The foramen magnum was decompressed with a micro-technique using a high-speed drill and a rongeur (Figure 4). The bony part of the bifid lamina and occipital bone were extremely soft. The bifid arch of the C1 vertebra was covered with the fibrous band which was cut as a part of posterior decompression. An adequate decompression of CCJ and regular closure of the wound was obtained.

The postoperative recovery went uneventfully, and the patient was released from hospital one week after surgery.

Follow-up MRI scan of cranio-cervical junction after three months and one year revealed residual stenosis of CCJ and myelopathy of cervical medulla. Nevertheless, the postoperative diameter of decompressed CCJ was insignificantly wider in the sagittal and axial planes compared to preoperative MRI scan (Figure 5).

## 2. Discussion

Saul-Wilson syndrome is considered one of the rarest diseases worldwide with only fourteen known cases from the 1990s until the present. Saul and Wilson were the first to describe the syndrome, which was presented as severely short stature, craniofacial dysmorphism and microcephaly [3]. Clinical appearance in our patient primarily corresponded and overlapped with Berardinelli-Seip syndrome (BSS) which was excluded after a genetic exam.

In the scope of the presented case conditioned by congenital disorders, further surgical approaches should be avoided, and further treatment should be focused merely on symptomatic treatment under guidance of paediatricians.

A preoperative magnetic resonance imaging (MRI) scan revealed a severe myelopathy at the level of cranio-cervical junction and radiologic signs of atlanto-axial instability. Therefore, the authors decided to obtain only bone decompression of the foramen magnum and C1 vertebra to prevent any possible neurological deterioration with the aim of achieving a better radiologic outcome, e.g., partial or nearly complete regression of myelopathy. There were no radiological signs of platybasia and basilar invagination according to radiographic measurements of cranio-cervical junction.

Despite an adequate decompression of the CCJ, a postoperative follow-up MRI scan revealed only a limited widening of the diameter of the cranio-cervical junction caused by radiologically confirmed atlanto-axial instability. There was no sign of regression of myelopathy. The authors did not confirm the utmost benefit of surgery according to radiological follow-up.

Ferreira et al. compared and followed-up characteristics of their patients which revealed short stature and progeroid appearance in all patients. A significant part of SWS clinical findings are skeletal abnormalities, which are described as osteoarticular pain followed by decrease range of motion. Bone fragility and poor bone healing are also presented as secondary common clinical features in patients suffered from SWS [4]. According to the aforementioned bone characteristics, skeletal surgery in these patients should be considered only as the last possible treatment option. Physical attributes in our patient corresponded to the aforementioned bone characteristics; therefore, any surgical instrumentation with the aim of stabilization of CCJ was impossible to be achieved. An attempt at hardware placement at all costs could have resulted in further damage and additional devastating instability of the CCJ. From this stance, our surgery can only be considered as an attempt to resolve myelopathy.

Despite satisfying postoperative recovery, the main obstacle in further treatment of our patient remained his established clinical status in the scope of his primary disease and radiologic confirmation of persisting myelopathy. Another significant difficulty in the treatment approach was the impossibility of obtaining a prompt genetical confirmation of SWS, since it currently cannot be performed in the Republic of Croatia. A genetic exam was necessary to obtain an adequate diagnosis due to the physical features and physical development of the patient. 

Goel proposed a classification of atlantoaxial instability based on analysis of numerous subjects that included instabilities of all spinal segments. A novel classification defines types of central atlantoaxial instability (CAAD) with emphasis on the possibility of atlantoaxial instability without radiologic confirmation or influence of bone deformity. There are two types of atlantoaxial instability that define dislocation of the atlas in the anterior or posterior direction (types 1 and 2), and a third type that defines normal anatomic alignment with clinical and radiological evidence during the surgical procedure (type 3). Furthermore, Goel described CAAD as an independent clinical entity and the cause of cervical myelopathy, especially in abnormal conditions of cervical spine, included bifid arch of the atlas which is comparable to our case. Musculoskeletal and neurologic alterations as the most common hallmarks in patients with CAAD are not considered as embryological disorders. Contrary to this claim, based on a large number of subjects, musculoskeletal and neurological disorders in our patient have to be considered as a result of primordial growth disturbance as a part of SWS [5].

We were unable to surgically treat our patient properly due to his physical status, which included a very low bodily height and weight, shortened neck and soft bones, creating difficulties in achieving the usual operative approach. At the moment, the treatment of SWS remains mainly in the reach of paediatricians according to the clinical symptoms of the patient, mostly systemic and metabolic conditions.

In conclusion, the authors were confronted with an unusual condition in an extremely rare congenital disease. Due to its rarity, a possible preoperative radiologic confirmation of critical stenosis of CCJ in such cases should be carefully evaluated. If required, surgery should only be considered as a palliative treatment.

## Figures and Tables

**Figure 1 children-09-00532-f001:**
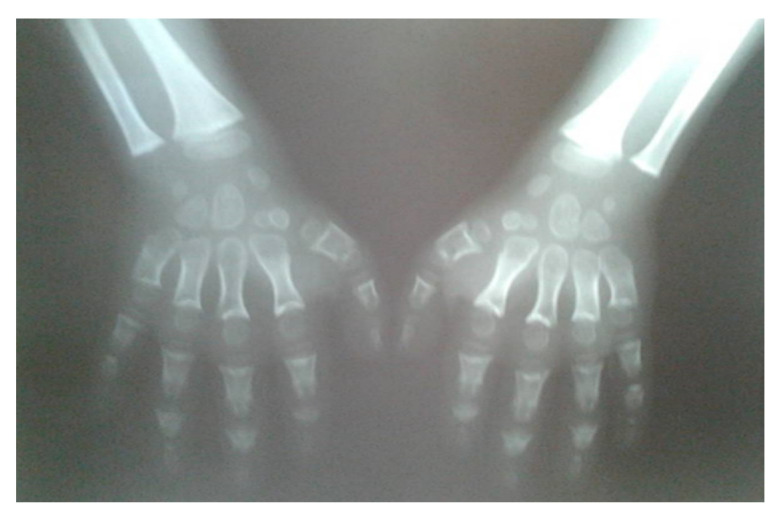
Radiographs of hands (10 years of age) revealed brachydactyly; wide and shortened phalanges of the fingers, cone-shaped epiphyses of phalanges, short metacarpals.

**Figure 2 children-09-00532-f002:**
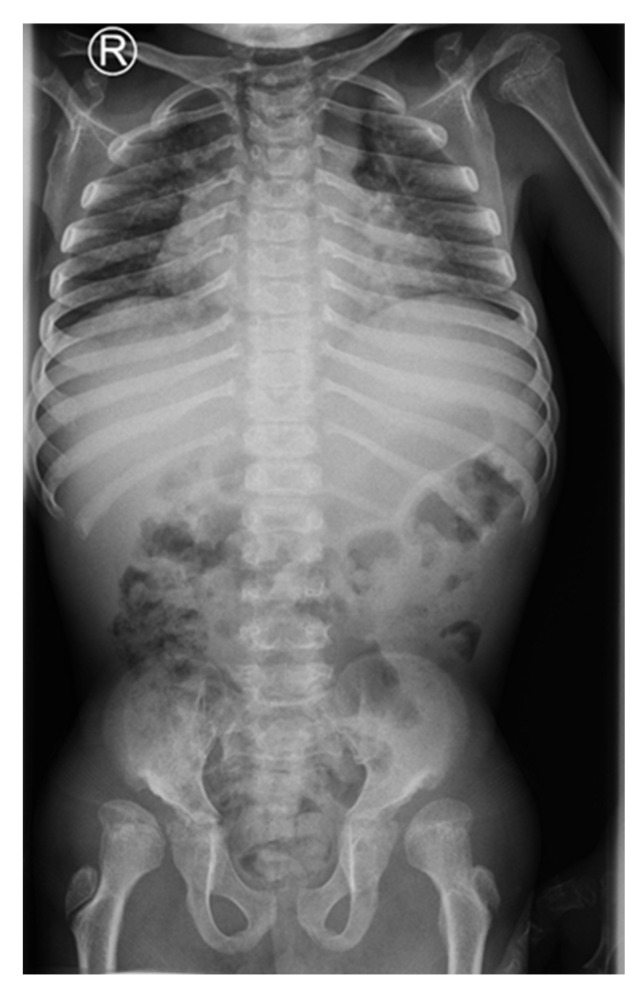
Dysplastic changes of the acetabulum, coxa valga, lucencies of the proximal femora, flattened vertebral bodies (platyspondyly), 11 years of age.

**Figure 3 children-09-00532-f003:**
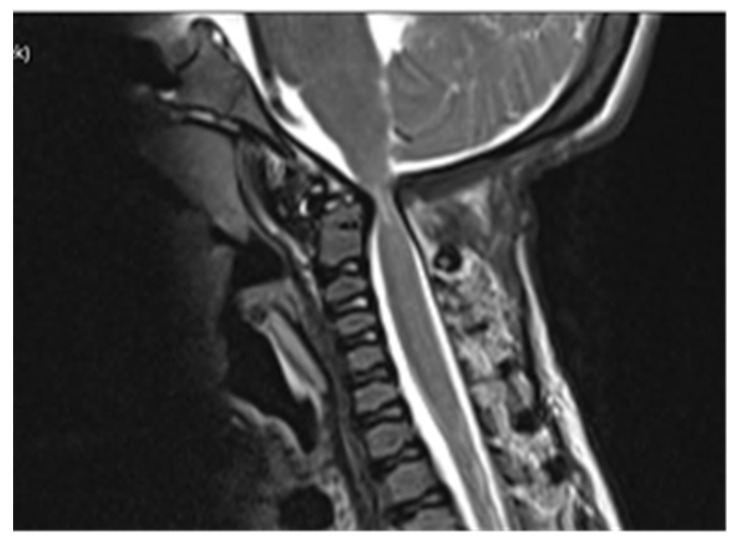
Preoperative MRI scanning of cranio-cervical junction revealed critical stenosis and myelopathy, 14 years of age.

**Figure 4 children-09-00532-f004:**
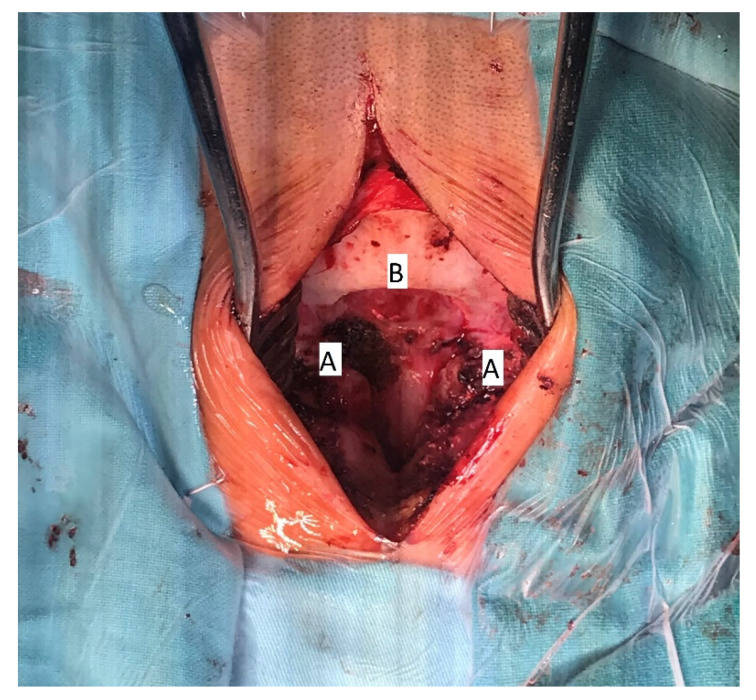
Intraoperative image of decompressive craniotomy (**B**); bifid arch of the atlas (**A**), 15 years of age.

**Figure 5 children-09-00532-f005:**
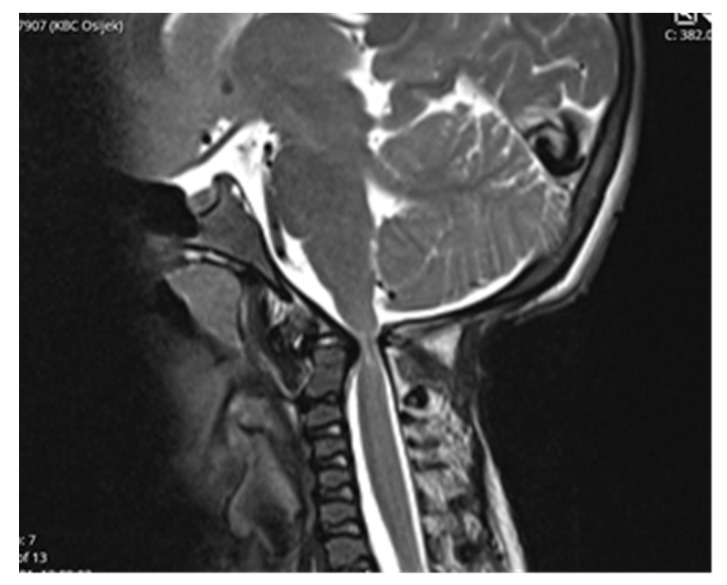
Postoperative MRI revealed insignificant radiological differences comparing to preoperative scanning with persistent myelopathy (15 years of age).

## Data Availability

Not applicable.

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
