# Peer review of "Severe Cranio-Cervical Stenosis in a Child with Saul-Wilson Syndrome: A Case Report"

_children, 2022, doi:10.3390/children9040532_

Round 1

Reviewer 1 Report

Dear authors, your case repot is well written and the topic is very interesting. The background information about the genetics is well done and understandable. Also the clinical description of the patients.

A few points could be improved:

the xray and MRI and CT scan should include age of the patient (days/months/years).

Your discussion is good but I doubt that you need to repeat the genetics and the Golgi system- instead of that I would start with the result of your case report to underline the main message

In the scope of the presented case conditioned by congenital disorders, further surgical approaches should be avoided and further treatment should be focused merely on symptomatic treatment under guidance of paediatricians. 

As this is really important for the reader! This message is important to understand how to decide in further cases. I would suggest that you should discuss on this problem more in your discussion part. What was your indication? Based on what (literature). What did you expect? Is the result (no significant improvement) also known in other rare spinal disorders (Literature) with a similar message: no surgery based on MRI scan as it is no classic Atlanto- axillar instability.

In general, a very interesting case- report with an important message!

Author Response

Responses to Reviewer 1 comments

Dear reviewer 1, thank you very much for considering our article.

I would like to answer to your suggested points:

Reviewer 1, Point 1: A few points could be improved: the xray and MRI and CT scan should include age of the patient (days/months/years).

Response 1: The X-ray, MRI and CT scans will include suggested terms.

Reviewer 1, Point 2: Your discussion is good but I doubt that you need to repeat the genetics and the Golgi system- instead of that I would start with the result of your case report to underline the main message

Response 2: We will exclude the part about genetics from discussion as you suggested.

Reviewer 1, Point 3: What was your indication? Based on what (literature). What did you expect? Is the result (no significant improvement) also known in other rare spinal disorders (Literature) with a similar message: no surgery based on MRI scan as it is no classic atlanto-axillar instability.

Response 3: Medulopathy and cranio-cervical instability were the main indications for surgery confirmed by MRI. Walking disability was already established in our patient due to skeletal dysplasia, no other neurological disorders were noted. Similar cases of achondroplasia and surgical techniques used were recently published by Antunes et al. (Decompressive Surgery for Craniovertebral Foramen Magnum Stenosis with Medullary Compression in Paediatric Skeletal Dysplasia Syndromes ), therefore their results were evident due to neurologic disorders in their patients comparing to our patient. Besides similar preoperative MRI scans and the same surgical procedure comparing their cases with our case, we expected at least better radiologic outcome with possibility of partially or nearly complete regression of myelopathy. 

The other article describing decompression of cranio-cervical junction (CCJ) was Posterior Decompression of the Craniovertebral Junction in Syringomyelia Combined with Chiari-1 Malformation In Children by Sanakoeva et al. where they describe the possibilities of CCJ decompression and its effect in postoperative regression of syringomyelia in almost 80% of their cases – we also mention that article only due to the same surgical procedure.  

Prior to our surgery I have discussed our case with prof. Goel who recommended CCJ stabilization which was our main goal to prevent further CCJ instability. In conclusion, our patient was 15-years old, only one meter tall / long and 15 kgs of body weight. His physical attributes did not allow us to even place him at the surgical table, also his bones were completely underdeveloped, soft and thin and any surgical instrumentation and stabilization was impossible to achieve, therefore we tried decompression as the only possible resort. An attempt of stabilization at all costs could have resulted with further damage of CCJ.

I hope you will accept our reply.

Kind regards,

Nenad Koruga

Reviewer 2 Report

The paper presented for the review is aimed to present a rare case of a boy suffering from Saul-Wilson syndrome complicated with critical stenosis of cranio-cervical junction (CCJ) and atlanto-axial dislocation.
The SWS syndrome is a rare skeletal dysplasia with multisystemic involvement. The reviewer observed personally three patients with genetically confirmed Saul-Wilson syndrome, all of them had spinal problems including cervical instability and stenosis.
The presented case generally demonstrates the reported previously in the literature features of the spinal involvement characteristic for the SWS. The authors postulate critical character of the stenosis and myelopathy. Nevertheless they don’t present neither clinical (detailed description of the neurological symptoms) nor instrumental (ENMG, evoked potentials) data.
According to the description of the surgical procedure, it included partial resection of the posterior arch of the atlas and widening of the foramen magnum. As the authors say, their procedure did not reach neither clinical nor radiological results. In the discussion the authors cite the paper of Goel and present his classification of the atlantoaxial instability. In his original paper Goel clearly defines three types of instability which also mentioned by the authors of the present paper. The quality of the figures does not give any chance to properly classify the instability of the described patient according to the Goel’s classification. But it looks more compatible to the type 1 instability in which not the posterior decompression but anterior stabilization is needed. 
The other figures illustrating the paper are of extremely poor quality which does not give a chance to distinguish their important details at all (e.g. radiograph of the feet) or partially (the rest radiographs). The quality of their English text is far from perfection even for the non-native speaker and should be considerably upgraded before publication.
Concluding the review, The paper needs major revision and resubmission.

Author Response

Responses to Reviewer 2 comments

Dear reviewer 2, thank you very much for considering our article.

I would like to answer to your suggested points:

Reviewer 2, Point 1: The reviewer observed personally three patients with genetically confirmed Saul-Wilson syndrome, all of them had spinal problems including cervical instability and stenosis.

Response 1: We observed and surgically treated only one patient with cervical instability and stenosis (not three as mentioned in comments and suggestions).

Reviewer 2, Point 2: The authors postulate critical character of the stenosis and myelopathy. Nevertheless they don’t present neither clinical (detailed description of the neurological symptoms) nor instrumental (ENMG, evoked potentials) data.
According to the description of the surgical procedure, it included partial resection of the posterior arch of the atlas and widening of the foramen magnum. As the authors say, their procedure did not reach neither clinical nor radiological results. In the discussion the authors cite the paper of Goel and present his classification of the atlantoaxial instability. In his original paper Goel clearly defines three types of instability which also mentioned by the authors of the present paper. The quality of the figures does not give any chance to properly classify the instability of the described patient according to the Goel’s classification. But it looks more compatible to the type 1 instability in which not the posterior decompression but anterior stabilization is needed. 

Response 2: Medulopathy and cranio-cervical instability were the main indications for surgery confirmed by MRI. Walking disability was already established in our patient due to skeletal dysplasia, other neurological symptoms were exluded. Similar cases and surgical techniques were recently published by Antunes et al. (Decompressive Surgery for Craniovertebral Foramen Magnum Stenosis with Medullary Compression in Paediatric Skeletal Dysplasia Syndromes ), therefore their results were evident due to neurological symptoms of their patients. Besides similar preoperative MRI scans and the same surgical procedure comparing their cases with our case, we expected at least better radiologic outcome with possibility of partially or nearly complete regression of myelopathy, from that stance EMNG or evoked potentials were excluded.

Prior to our surgery I have discussed our case with prof. Goel who recommended CCJ stabilization which was our main goal to prevent further CCJ instability. In conclusion, our patient was 15-years old, only one meter tall / long and 15 kgs of body weight. His physical attributes did not allow us to even place him at the surgical table, also his bones were completely underdeveloped, soft and thin and any surgical instrumentation and stabilization was impossible to achieve, therefore we tried decompression as the only possible resort. An attempt of stabilization at all costs could have resulted with further damage of CCJ.

In your comments you mentioned Goel's classification and type 1 instability which is needed to be stabilized anteriorly. In such cases of cranio-cervical instability anterior stabilization is never an option, even Goel in his article does not mention such possibility.

Also, according to Goel from the same article: „CAAD as a discrete clinical entity - CAAD can be an independent clinical entity and a cause of cervical myelopathy. High degree of suspicion that is based on clinical findings and radiological observations can guide towards the correct diagnosis. Presence of unusual cervical lordotic curvature, bone fusions, bifid arch of atlas and other telltale evidences can guide the surgeon towards the diagnosis of CAAD. Identification and treatment of CAAD can lead to relief from disabling symptoms.”

According to all previously mentioned physical obstacles in our patient, we were unable to observe and properly treat our patient as a genuine CAAD patient due to his primary disease. 

Reviewer 2, Point 3: The other figures illustrating the paper are of extremely poor quality which does not give a chance to distinguish their important details at all (e.g. radiograph of the feet) or partially (the rest radiographs).

Response 3: Figures we used in our article were taken from the patient's archive data using the phone camera. We will try to make the images of better quality. Also, all other figures besides MRI and CCJ were taken in order to reveal clinical disturbances of our patient as Markova et al. did in their article: Clinical and genetic characteristics and orthopedic manifestations of the Saul–Wilson syndrome in two Russian patients. We can exclude it if you consider these images unnecessary and include intraoperative figure of CCJ decompression.

Reviewer 2, Point 4: The quality of their English text is far from perfection even for the non-native speaker and should be considerably upgraded before publication.

Response 4: Spell and grammar check of our article were thoroughly checked by professor of English, even though we agreed that our manuscript should undergo English revision by MDPI.

I hope you will accept our reply.

Kind regards,

Nenad Koruga

Round 2

Reviewer 2 Report

The paper presented for the second round of the reviewing. The authors did some minor modifications according to the previous recommendations. But there are still major problems with the presented material according to the reviewer’s point of view. In the first review it was mentioned that the reviewer (not the authors) has 3 patients with SWS under surveillance – just to notice that I have personal experience with this condition. The authors mentioned that the reason for their surgery was the presence of the myelopathy. It should be pointed out again that myelopathy is not a radiological phenomenon but the clinical diagnosis (Seidenwurm DJ; Expert Panel on Neurologic Imaging. Myelopathy. AJNR Am J Neuroradiol. 2008 May;29(5):1032-4. PMID: 18477657; PMCID: PMC8128582). The authors still did not present sufficient clinical and neurophysiological data regarding the diagnostic criteria of the myelopathy but just speculate regarding the MRI data. The main problem is the surgical approach itself. As the authors mentioned in their response to the reviewer, their personal communication with the doctor Goel (the author of the classification discussed in the presented paper) according to his classification the surgery should address not the posterior, but anterior structures first of all. The authors decided to follow their first plan with posterior decompression. That is why it’s not surprising that their results were unsatisfactory. In fact, this is should be the main message of the paper: inadequate choice of the treatment modality leads to insufficient results. Otherwise it remains unclear why the authors did not reach the main goal of the surgery. But from my point of view the answer is clear. Regarding the figures, the quality of the radiographs is in the presented version is a little better .but still rather poor – the decision on that is on the editor. Bu the way the CT scan of the craniovertebral region demonstrates only bifid posterior arch of the atlas which is not a part of the spectrum of the pathological changes characteristic for the described type of the stenosis. The paper still needs major conceptual reworking.

Author Response

Response to Reviewer 2 comments:

Point 1: The authors mentioned that the reason for their surgery was the presence of the myelopathy. It should be pointed out again that myelopathy is not a radiological phenomenon but the clinical diagnosis…

Response 1: Primary aim of our surgery was cranio-cervical instability with critical stenosis; myelopathy was a result of instability and consequent compression of medulla, therefore further instability would lead to potential aggravating of neurological condition in our patient, sooner rather than later. 

Point 2: The authors still did not present sufficient clinical and neurophysiological data regarding the diagnostic criteria of the myelopathy but just speculate regarding the MRI data.

Response 2: Clinical data of our patient is described in our manuscript, no significant neurological deficit were found. Therefore neurophysiological data were not considered during the preoperative period. Walking disability and symptoms are results of patient's primary disease; authors would not gain any diagnostic or clinical insight using evoked potentials or EMG. Taking into account this claim, Goel et al. in their article “Central or Axial Atlantoaxial Dislocation as a Cause of Cervical Myelopathy: A Report of Outcome of 5 Cases Treated by Atlantoaxial Stabilization” did not use any of these tests despite clear neurologic deficits in their patients. Also, the same can be found in (https://doi.org/10.1007/s10143-018-01070-4) by Henderson et al.

Point 3: The main problem is the surgical approach itself. As the authors mentioned in their response to the reviewer, their personal communication with the doctor Goel (the author of the classification discussed in the presented paper) according to his classification the surgery should address not the posterior, but anterior structures first of all. The authors decided to follow their first plan with posterior decompression. That is why it’s not surprising that their results were unsatisfactory. In fact, this is should be the main message of the paper: inadequate choice of the treatment modality leads to insufficient results. 

Response 3: Authors strongly disagree with your comment. As already mentioned in the first round of reviewing, posterior approach is the ONLY surgical approach no matter what type of instability! There is no point to reiterate it again.

Point 4: Otherwise it remains unclear why the authors did not reach the main goal of the surgery. But from my point of view the answer is clear.

Response 4: The answer is clear because we already answered that question after the first round.

Point 5: Regarding the figures, the quality of the radiographs is in the presented version is a little better .but still rather poor – the decision on that is on the editor. Bu the way the CT scan of the craniovertebral region demonstrates only bifid posterior arch of the atlas which is not a part of the spectrum of the pathological changes characteristic for the described type of the stenosis.

Response 5: Figures were taken directly from the hospital’s radiological service, we cannot achieve better quality. We excluded CT image of bifid arch of the atlas.

Point 6: The paper still needs major conceptual reworking.

Response 6: Authors agreed that there is no room for major reworking, also the other reviewer did not ask for further reworking after the first round. We would like to thank you for your time.